# Dietary Flavonoid Intakes in France Are Linked to Brewed Tea Consumption and to Socioeconomic Status: Analyses of the Third French Individual and National Food Consumption (INCA3) Survey for Children and Adults

**DOI:** 10.3390/nu16081118

**Published:** 2024-04-10

**Authors:** Florent Vieux, Matthieu Maillot, Adam Drewnowski

**Affiliations:** 1MS-Nutrition, 27 bld Jean Moulin Faculté de Médecine la Timone, Laboratoire C2VN, CEDEX 5, 13385 Marseille, France; florent.vieux@ms-nutrition.com (F.V.); matthieu.maillot@ms-nutrition.com (M.M.); 2Center for Public Health Nutrition, University of Washington, Box 353410, Seattle, WA 98195, USA

**Keywords:** flavonoids, flavan-3-ols, brewed tea, demographics, socioeconomic status, Third French Individual and National Food Consumption Survey (INCA3), children, adults

## Abstract

Flavonoids from green and black tea may benefit cardiovascular health. Brewed tea consumption and flavonoid intake in France have not been previously explored. This study assessed the dietary intake of flavonoids among French children and adults, using 3 days’ dietary recall for 3896 persons aged >4 y in the Third French Individual and National Food Consumption Survey (INCA3). Foods consumed by INCA 3 participants were manually matched with the flavonoid content of foods from the French PhenolExplorer database and the US Department of Agriculture expanded flavonoid database (2018 version). The six subclasses of flavonoids were flavan-3-ols, flavanones, anthocyanidins, flavonols, flavones, and isoflavones. Flavonoid intake was stratified by age subgroups (children and adults separately) and examined using socio-demographics and tea consumption patterns. Mean flavonoid intake was 210 mg/d. Flavonoids in the French diet were predominantly flavan-3-ols (147 mg/d), of which tea is the main source. The effects of age, education, income, and socio-professional category (SPC) on flavonoid intake were all significant (*p* < 0.0001). Brewed tea consumers were 31.88% of French adults and 3.79% of children. Brewed tea consumption and flavonoid intake were highly correlated. The highest brewed tea and flavonoid intakes were found among individuals with the highest SPC and education levels. Flavonoid intake in France was associated with brewed tea consumption and with higher education and income.

## 1. Introduction

Brewed tea is a major source of dietary flavonoids [1], functional compounds with beneficial health outcomes [2,3,4,5]. Dietary flavonoids are normally divided into subclasses: flavan-3-ols (tea), flavanones (citrus fruit and juices), anthocyanidins (berries), flavonols (tea, onions, and potatoes), flavones (tea, and peppers), and isoflavones (soy) [6]. Tea catechins are a subset of monomeric flavan-3-ols that occur in green tea rather than in fermented black tea [3]. Flavonoids, and particularly flavan-3-ols and anthocyanidins subclasses, have been associated with a reduction in the risk of cardiovascular diseases [4].

Analyses of nationally representative dietary surveys in the US (NHANES 2007-10) [2] showed that 78% of total flavonoids were provided by tea. Tea (mostly black) accounted for 94% of flavan-3-ols, 37% of flavonols, and 10% of flavones in the US diet. Orange juice and oranges provided flavanones, whereas blueberries were the major source of anthocyanidins, followed by wine [2]. Flavones came from mixed dishes containing onions, peppers, and potatoes, whereas isoflavones came from soy products [2].

Given the overwhelming contribution of fermented black tea to flavonoid intake among US adults, many of the health benefits of flavonoids have been attributed to tea consumption [1,7]. However, both tea consumption and flavonoid intake in the US were strongly associated with socioeconomic status (SES) [1,8], a well-known determinant of health outcomes. Most likely to consume tea in the US were older adults, persons with higher education and incomes [8], and those with more active lifestyles [2]. In general, tea consumption among US children was low [8].

By contrast, little is known about tea consumption in relation to flavonoid intake and SES variables in France. Data on socio-demographic characteristics of French tea consumers are very limited. The characterization of flavonoid intake in Frace by amount, flavonoid class, and by the socioeconomic status of the consumers would be a valuable addition to the literature. Tea is not the only item that follows a socioeconomic gradient [1]. In the US, the consumption of whole fruit, including anthocyanin-rich berries, has been associated with higher SES [1], whereas the consumption of 100% orange juice was associated with lower SES [9]. These gradients may affect diet quality and the distribution of health outcomes.

Estimating the amounts of flavonoids in the French diet requires a nationally representative dietary intake database joined with a nutrient composition database that contains the flavonoids of interest. We were able to join the most recent INCA 3 dietary surveys [10] with flavonoid data from the French PhenolExplorer [11,12,13] and from the USDA’s Expanded Flavonoid Database [14,15,16,17]. This allowed us to compare flavonoid intake by socio-demographics and identify their main sources in foods. Links to databases are provided in the data availability statement. Additional analyses were conducted on percentile distributions of intake for flavonoid subclasses [1]. Finally, flavonoid intakes of tea consumers and non-consumers were compared.

## 2. Materials and Methods

An overview of the steps taken to generate the final analytical sample for the study is shown in Figure 1.

### 2.1. Multiple 24 h Dietary Recalls

Dietary intake data came from the nationally representative INCA 3 dietary survey of the French population conducted by the French Agency for Food, Environmental and Occupational Health and Safety (ANSES) in mainland France between February 2014 and September 2015 [10]. Self-reported dietary intakes were collected using 3 non-consecutive 24 h dietary recalls for 2 weekdays and 1 weekend day spread over 3 weeks. After the exclusion of young children (<4 y), the final INCA 3 sample was 2121 adults (≥18 y) and 1775 children (≥4 y).

The methodology of dietary intake assessment through telephone interviews was adjusted by age group. Individuals aged 15 to 79 y were asked about their food and beverage consumption during the previous 24 h. However, they were not informed in advance of the days on which they would be interviewed, to avoid bias. Individuals aged <15 y were provided with food records and knew the day of data collection ahead of time. Data collection days were determined at the time of the home visit by the investigator. Children could be assisted by caregivers or other persons responsible for their nutrition (parents, grandparents, nanny, nursery care staff, or school). A follow-up call was made the day before the recording days and a phone interview call was most often carried out the next day, or at the latest 3 days later, to collect and complete the consumption record in the logbook. All interviews were conducted by telephone, using standardized updated French-specific version of GloboDiet software (formerly EPIC-Soft) [18], by professional interviewers specifically trained in the methods implemented and in the use of the software.

### 2.2. Participant Characteristics

Socio-demographic information collected for each individual included age group, socio-professional category (SPC), level of education, and total monthly income of the household per capita (consumption unit or CU in France). Among adults, three age classes were provided by INCA3 (18–45 y; 46–64 y; and ≥65 y). Following the International Standard Classification of Occupations [19], SPC was divided into “low” (mainly office and manual workers), “medium” (mainly craftspeople, company directors/owners and other intermediate professions), and “high” (mainly executives and self-employed professionals), and a fourth class, labelled “not working”, including retired, used to work, students and housewives/househusbands. Level of education was divided into 4 classes: “primary and middle school” (<12 y), “high school” (12 y), “1 to 3 years of post-secondary education” (12–16 y), and “4 or more years of post-secondary education” (>16 y). Income per CU was divided into 5 classes: “<900 EUR/month/CU”, “[900–1340] EUR/month/CU”, “[1340–1850] EUR/month/CU”, “>=1850 EUR/month/CU”, and “Unknown”. Body mass index (BMI) was divided into “Thin”, “Normal”, “Overweight”, “Obesity”, “Morbid obesity”.

### 2.3. Consumption of Tea and Beverages

Tea consumers were classified as those individuals who consumed regular or decaffeinated brewed tea (black or green) on at least one occasion during the 3-day data collection period. Herbal teas (e.g., chamomile) and other non-tea infusions such as Rooibos were assigned to the “Coffee and herbal teas” category (Appendix A). Pre-sweetened teas (bottled or canned) were placed in the “carbonated soft drinks” category.

### 2.4. Flavonoid Databases

The PhenolExplorer database [11,12,13] provides the contents of 500 different polyphenols for 458 foods. The flavonoids were grouped into subclasses: flavan-3-ols, flavanones, flavonols, flavones, anthocyanins, chalcones, dihydrochalcones, dihydroflavonols, and isoflavones.

The USDA Expanded Flavonoid Database [14,15,16,17] contains analytical values for 25 flavonoid compounds for 512 food items. Flavonoid subclasses were flavan-3-ols, flavanones, flavonols, flavones, and anthocyanidins (aglycones).

Food groups and food items containing flavonoids were identified mostly in Phenol Explorer, with some coming from the USDA Flavonoid databases. Individual food items from flavonoid-containing food groups in INCA3 were then manually matched to their flavonoid content based on product description. Out of 2835 unique foods consumed by the INCA 3 population sample, 1917 were in flavonoid-containing food groups. Out of these, 355 had matching flavonoid content in the PhenolExplorer (91% of the matches) or the USDA databases (9% of the matches).

This matching allowed us to estimate daily intakes of total flavonoids and subclasses, anthocyanidins, flavanones, flavonols, flavones isoflavones, and flavan-3-ols, for all participants in the INCA3 survey.

### 2.5. Data Availability and Ethical Approval

The INCA3 study was conducted according to the guidelines laid down in the Declaration of Helsinki and the study protocol obtained authorization from the CNIL (National Commission on Informatics and Liberty) on 2 May 2013 (Decision DR 2013-228), after approval by the CCTIRS (Advisory Committee on Information Processing in Health Research) on 30 January 2013 (notice 13.055). Verbal informed consent was obtained from all subjects. Verbal consent was witnessed and formally recorded. All INCA3 data are publicly available on websites [20].

### 2.6. Statistical Analyses

The survey-weighted mean intakes of tea, total flavonoids, and flavonoid classes were evaluated separately for children and adults. The distribution of total flavonoid intakes (5, 15, 25, 50, 75, 85, and 95-ile cut points) was estimated among children and adults. For adults, means intakes and percentile distribution comparisons were made by sex, age group, occupation (SPC), income (ICU), and education. Food group sources of flavonoids and flavonoid classes were also identified separately for children and adults. Flavonoids and subclass intakes were compared between adult tea consumers and non-consumers. Subsequent analyses, conducted among tea consumers only, tested the association between tea consumption (in g/d) and total flavonoid intake as well as subclasses of flavonoids intake (in mg/d).

All analyses were conducted using SAS software version 9.4 (SAS Institute Inc., Cary, NC, USA). Survey-weighted means and survey-weighted percentages were estimated using the SURVEYMEANS and SURVEYFREQ procedures, respectively. Differences in continuous variables were tested using regression analysis with the SURVEYREG procedure. The threshold for statistical significance was *p* < 0.05.

## 3. Results

### 3.1. Tea Consumption and Flavonoid Intake by Socio-Demographics

Tea consumers, defined as those INCA 3 participants who drank brewed tea at least once during the 3 days, were 31.88% of adults and only 3.79% of children.

Figure 2 shows total flavonoid intake (in mg/d) in adults and children. In adults, the unadjusted mean intake was 210 mg/d for total flavonoids, of which there was 147 mg/d for total flavan-3-ols, 29 mg/d for flavanones, 15 mg/d for flavonols, 15 mg/d for anthocyanidins, 3 mg/d for flavones, and 1 mg/d for isoflavones. In children, the unadjusted mean flavonoid intake was 97 mg/d for total flavonoids, 46 mg/d for total flavan-3-ols, 38 mg/d for flavanones, 5 mg/d for flavonols, 5 mg/d for anthocyanidins, 3 mg/d for flavones, and <1 mg/d for isoflavones.

The patterns of tea consumption (in g/day) by sex, age group, SPC, ICU, education, and BMI are shown in Table 1. Data are means and standard deviations. Consistent with past studies, more tea was consumed by women than by men (141 vs. 70 g/day) and by adults with higher SPC and higher education (*p* < 0.001 for all). Tea consumption decreased with increasing BMI. Among adults, there were no significant effects of age group. Children consumed relatively little brewed tea (6 g/d). Table 1 also shows the total flavonoid intake (in mg/d), estimated at 206 mg/d for men and 214 mg/d for women. Analyses of the total flavonoid intake showed significant effects by age group, ICU, SPC, education, and BMI (*p* < 0.001 for all). Consistent with other studies, higher flavonoid intake was associated with older age, higher ICU, higher SPC, higher education, and lower BMI. Only the effect of sex was not significant.

Flavonoid intake by flavonoid class showed distinct socio-demographic and economic profiles. Females had a higher intake of flavan-3-ols (tea), whereas males had higher intakes of flavanones (citrus), anthocyanidins (berries), and flavones. The positive effect of higher age was observed for total flavonoids, flavan-3-ols, flavonols, flavones, and especially anthocyanins.

Strong income effects were observed for total flavonoids, flavan-3-ols (tea), flavonols, and anthocyanins (berries). There was an income effect for flavones, but the amounts were very low. No significant income effect was observed for flavanones and isoflavones. There were significant effects of SPC for total flavonoids and all flavonoid subclasses except isoflavones. No effect of education was observed for isoflavones, flavanones or flavones. Flavan-3-ols and flavonols decreased with increasing BMI, while an inverse U shape was observed for anthocyanins. Others subclasses of flavonoids did not differ between BMI classes.

### 3.2. Distribution of Flavonoid Intake

Table 2 shows the percentile distribution of total flavonoid intake by INCA3 socio-demographics and BMI. Shown are values for 5, 15, 25, 50 (median), 75, 85, and 95th percentiles. For adults, the population distribution of total flavonoid intake was 58 mg/d (1st quartile), 139 mg/d (median), and 299 mg/d (3rd quartile). The 95th percentile cut-point was 635 mg/d. For children, the population distribution of total flavonoid intake was 34 mg/d (1st quartile), 70 mg/d (median), and 127 mg/d (3rd quartile). The 95th percentile cut-point was 269 mg/d. The distribution of tea flavonoid intakes among the whole sample and among tea consumers only were established for adults and children using 5, 15, 25, 50, 75, 85, and 95-ile cut points (Appendix A).

### 3.3. Food and Beverage Sources of Dietary Flavonoids

Figure 3 shows flavonoid intake by food sources in adults and children. The main sources for adults were tea, fruits, nuts and seeds, alcoholic beverages, fruit and vegetable juices, and chocolate confectionery. Children obtained total flavonoids from fruit and vegetable juices, fruits, nuts and seeds, dairy products and substitutes, and chocolate confectionery.

Table 3 shows flavonoid consumption by flavonoid class and food sources. Among adults, most flavonoids were provided by flavan-3-ols; that is to say, by brewed tea. Brewed tea and fruits, nuts, and seeds were the main sources of flavan-3-ols; fruit juices (citrus) provided flavanones and fruit (berries) was the source of anthocyanins. Alcoholic beverages such as wine were another source of flavonoids among French adults.

In children (Table 4), brewed tea was consumed in small amounts. Fruit juices, which were the main sources of flavonoids, provided flavanones. Fruits (contributor number 2) provided flavan-3-ols and anthocyanins.

### 3.4. Brewed Tea as the Main Source of Dietary Flavonoids

Table 5 shows the unadjusted mean flavonoid intake by adult brewed tea consumers and non-consumers. Statistical tests were performed without adjustment and after adjusting for sex, age group, ICU, SPC, education, and energy intake. Significant differences (*p* < 0.0001 for all) were obtained for total flavonoids, flavan-3-ols, and flavonols (all present in tea), for which tea consumers had higher significantly intakes than did non-consumers. Intakes of flavanones, anthocyanins, flavones, and isoflavones did not differ by tea consumption status except in unadjusted mean of isoflavones.

Figure 4 shows a scatterplot of tea consumption (in g/d) plotted against total flavonoid intake (in mg/d) and flavonoid subclasses in adult tea consumers. The positive correlation between tea consumption and total flavonoids was highly significant, with an R2 = 0.7968 (*p* < 0.0001).

For flavonoids subclasses, significant correlations (*p* < 0.0001 for all) were obtained for flavan-3-ols and flavonols (all present in tea) but not for flavanones (citrus), anthocyanins (berries), flavones (mixed dishes), and isoflavones (soy).

## 4. Discussion

The present analyses of the most recent French INCA3 dietary intake data merged with the French PhenolExplorer database allowed us to estimate total flavonoid intake in France at 210 mg/d for adults and 97.4 mg/d for children. The median flavonoid intake in France was 139 mg/d for adults and 70 mg/d for children. Adults derived most of their total flavonoids from brewed tea, followed by fruits and alcoholic beverages. Tea flavonoids (mostly flavan-3-ols) accounted for 82.8 mg/d (39% of the flavonoid intake). Men drank less tea than women and many have derived a larger proportion of dietary flavonoids from alcoholic beverages (wine and beer). Children consumed tea rarely and in small amounts. Most flavonoids in the French children’s diets came from fruit and fruit juices.

The present estimates for France are well below previous estimates for the US, which were also based on representative population samples [21,22,23]. Analyses of the 2011-2016 NHANES data [1] estimated unadjusted mean flavonoid intake among US adults at 256 mg/d for total flavonoids, 205 mg/d for total flavan-3-ols (of which 86 mg/d catechins), 12 mg/d for flavanones, 20 mg/d for flavonols, 16 mg/d for anthocyanidins, 1 mg/d for flavones, and 2 mg/d for isoflavones. Percentile cut-points for total flavonoids among adults were 27 mg/d (1st quartile), 72 mg/d (median), and 274 mg/d (3rd quartile). The 90th percentile cut-point was 690 mg/d.

Earlier analyses of the 2001-02 NHANES using the USDA Flavonoid Content of Selected Foods database, Release 2, estimated flavonoid intake at 132 mg/d, mostly from flavan-3-ols (112 mg/d) and flavonols (10 mg/d) [24]. Analyses of 2007–2010 NHANES by Sebastian et al. [2], using the more recent Flavonoid Values for the USDA Food Codes database, estimated adults’ (>20 y) flavonoid intake at 246 mg/d. Flavonoid intake based on a modeled French diet was estimated to be 247 mg/d, not including anthocyanins [25].

The present estimates for total flavonoid intake among French children were a mean of 97 mg/d and a median of 70 mg/d. By contrast, the unadjusted mean intake for older children (ages 9–19 y) in the US NHANES 2011-16 sample was 133 mg/d for total flavonoids, 100 mg/d for total flavan-3-ols (of which 45 mg/d catechins), 11 mg/d for flavanones, 11 mg/d for flavonols, 9 mg/d for anthocyanidins, and <1 mg/d for flavones and isoflavones [1]. The median value for total flavonoids in children was 43 mg/d. Lower mean values and higher median values observed in France might be explained by differences in food consumption patterns and different age groups (>9 y in the US).

There are no agreed upon guidelines for flavonoid intake. Based on a systematic review and meta-analyses of published data [26], there is moderate evidence to support cardiometabolic protection resulting from flavan-3-ol intake in the range of 400–600 mg/d. This guideline was based on beneficial effects across a range of disease biomarkers and endpoints that came from clinical trials [26]. The present intake of flavan-3-ols in France is in the order of 147 mg/d, lower than the estimates of 205 mg/d in the US. Flavan-3-ol intake in the order of 400–600 mg/d would correspond to the 90th percentile or above.

Brewed tea is the principal dietary source of flavonoids, and notably flavan-3-ols. In the US, adult tea consumers had a higher intake of total flavonoids (610 mg/d compared with 141 mg/d for non-consumers) and flavan-3-ols (542 mg/d compared with 97.8 mg/d for non-consumers) (*p* < 0.001 for both). French tea consumers also had a much higher flavonoid intake compared to tea non-consumers (385 mg/d vs. 128 mg/d).

There was a socioeconomic gradient in flavonoid intake both in France and in the US [23,24,27], though income effects in the US were not always consistent [27]. Consistent with the present results, the highest total flavonoid intake in the US was found in adults aged 51–70 y (293 mg/d), and in groups with college education (251 mg/d) and with higher incomes (IPR > 3.5:249 mg/d) (*p* < 0.001 for all). Not all studies of flavonoid intake were able to address intake patterns by sex, age, or socioeconomic status.

The current trends toward flavonoid-rich diets are justified, in part, by the cardioprotective nature of flavonoids. Flavonoids, and particularly flavan-3-ols and anthocyanidins, have been associated with a reduction in the risk of cardiovascular diseases [28,29,30], though multiple other health benefits have been reported as well [31,32,33]. The present analyses point to strong links between brewed tea consumption and flavonoid intake. Both in the US and in France, there were sharp differences in flavonoid intake between tea consumers and non-consumers. The present findings are also consistent with past reports that those French women who had higher flavonoid intake were older and better educated [28]. One report [27] found the mean flavonoid intake to be higher in women compared to men. That was not the case here; although French women consumed more tea than did men, men obtained more flavonoids from alcoholic beverages and the total amounts by gender were not significantly different.

The observed social gradient in tea consumption and flavonoid intake is worth noting because it is also observed in some flavonoid-related health outcomes. Interestingly, the same sharp gradient was found for anthocyanins both in France and the US. It is also worth noting that both tea consumption and flavonoid intake were inversely related to body mass index (and obesity prevalence), another infallible indicator of SES both in France and in the US.

Brewed tea is a major source of dietary flavonoids, predominantly flavan-3-ols. In France, only about one third of the population consumed brewed tea during the 2 days of INCA3 data collection. Their flavonoid intakes were higher compared to non-consumers. Increasing the consumption of unsweetened brewed tea may be one approach to obtaining more dietary flavonoids. There is also potential for further studies on coffee polyphenols, also listed in the PhenolExplorer database. Common polyphenols such as hydroxycinnamic acids, particularly chlorogenic acid, are highly bioactive compounds from coffee and berries.

The present study had limitations. INCA3 dietary intakes were based on self-report, where a potential underestimation of fluid intake is a recognized problem. The present analyses were based on three-day recalls which may not adequately capture habitual eating patterns. Because of the cross-sectional INCA3 study design, any potential associations between flavonoid intake and any health outcomes should be interpreted cautiously and no causal inferences should be made. Nonetheless, the scale and representativeness of INCA3 make it a valuable resource.

## 5. Conclusions

For the first time, flavonoid intake (and subclasses) has been estimated in a representative sample of French adults and children. Based on three-day food records, about 32% of adults but only 4% of children were classified as brewed tea consumers. Total flavonoid intakes were 210 mg/d in adults and 97 mg/d in children. Tea consumption was associated with significantly higher total flavonoid intake, and with a higher intake of flavan-3-ols, of which tea is the main source. Both tea consumption and flavonoid intake were associated with higher socio-economic status. Flavonoid intake in France was well below the recommended values.

## Figures and Tables

**Figure 1 nutrients-16-01118-f001:**
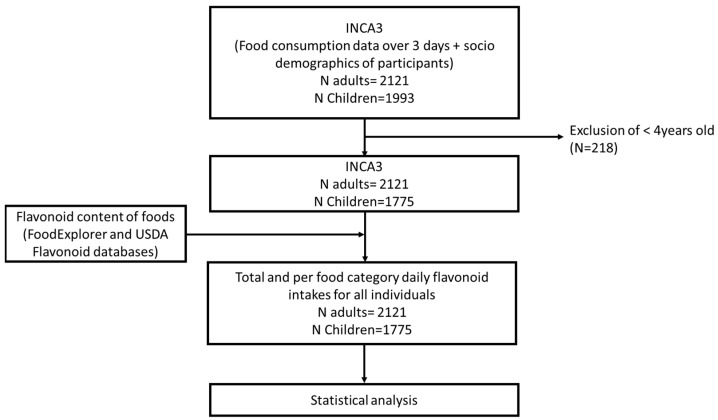
Overview of the methodology used to generate the study database.

**Figure 2 nutrients-16-01118-f002:**
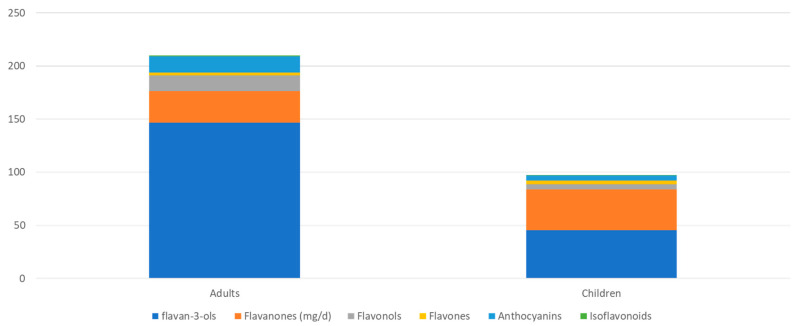
Total flavonoid intake (mg/d) in adults and children and contributions of flavonoid subclasses.

**Figure 3 nutrients-16-01118-f003:**
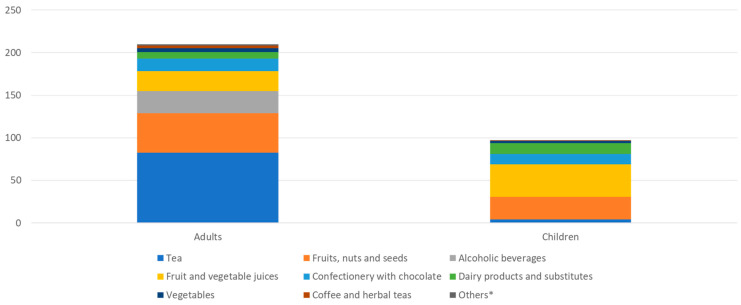
Total flavonoid intake (mg/d) in adults and children and food source contributions. * “Others” groups together all food sources which contributed to less than 1 mg/d in adults.

**Figure 4 nutrients-16-01118-f004:**
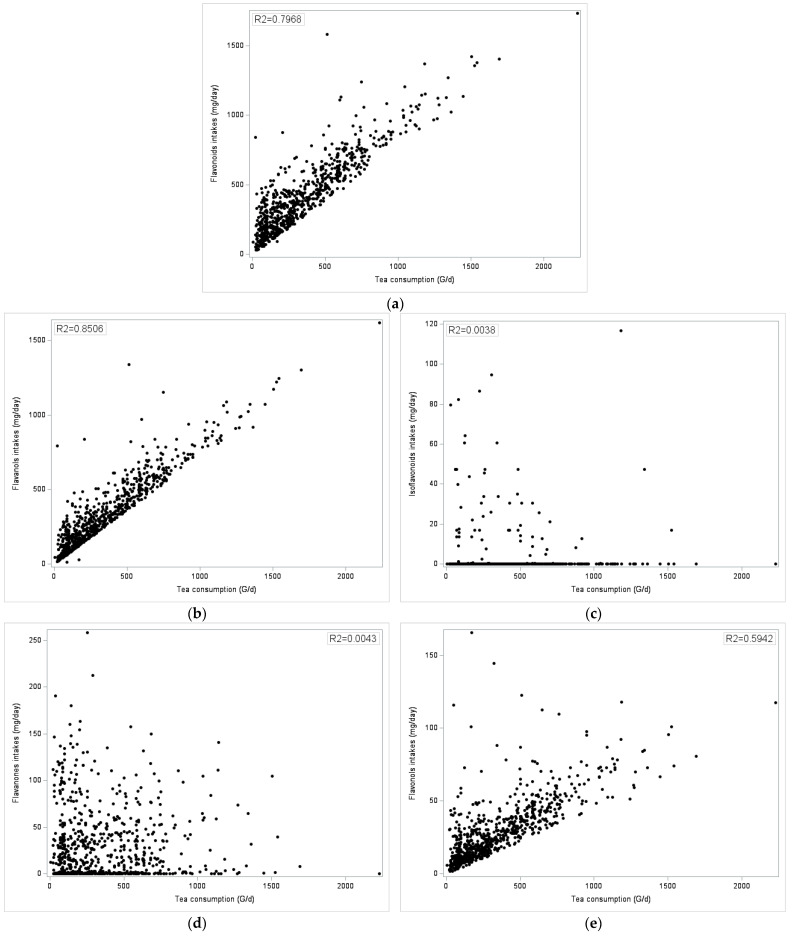
Scatterplot of tea consumption (g/day) versus total flavonoid intake and flavonoids subclasses in French adults (among tea consumers only, n = 758): (**a**) Total Flavonoids, mg/d, (**b**) Flavanols, mg/d, (**c**) Isoflavones, mg/d, (**d**) Flavanones, mg/d, (**e**) Flavonols, mg/d, (**f**) Flavones, mg/d, (**g**) Anthocyanins, mg/d. Total Flavonoids, Flavanols and Flavonols intakes correlated significantly with tea consumption.

**Table 1 nutrients-16-01118-t001:** Tea consumption (g/d) and flavonoid intake (mg/d) by sex, age group, ICU, SPC, education, and BMI (kg/m^2^) in adults, and tea consumption (g/d) and flavonoid intake (mg/d) in children. Data are means and standard deviations.

			Tea (g/d)		Total Flavonoids (mg/d)		Flavan-3-ols (mg/d)		Flavanones (mg/d)		Flavonols (mg/d)		Flavones (mg/d)		Anthocyanins (mg/d)		Isoflavones (mg/d)	
		N	Mean	SD	Mean	SD	Mean	SD	Mean	SD	Mean	SD	Mean	SD	Mean	SD	Mean	SD
Adults	All	2121	106.8	220.8	210.0	213.4	146.9	186.0	29.2	42.9	15.0	16.7	2.6	3.4	15.3	30.7	1.0	7.0
Gender	Men	887	70.1	176.7	205.5	195.1	133.6	163.2	32.9	49.3	14.8	15.9	2.8	3.7	20.6	36.9	0.8	7.2
	Women	1234	141.4	250.6	214.3	229.2	159.4	204.4	25.8	35.5	15.1	17.5	2.3	3.0	10.4	22.4	1.2	6.8
			*p* < 0.0001		*p* = 0.4619		*p* = 0.0139		*p* = 0.0064		*p* = 0.7088		*p* = 0.0093		*p* = 0001		*p* = 0.1804	
Age	18–44 y.o.	783	96.2	204.5	176.9	190.5	120.4	169.6	32.4	42.9	11.8	14.7	2.9	3.5	8.6	18.6	0.7	5.2
	45–64 y.o.	827	118.4	237.7	232.0	230.1	166.1	200.2	26.9	44.9	16.7	18.0	2.2	3.0	18.8	35.9	1.3	8.4
	65–79 y.o.	511	110.6	224.3	250.8	221.0	176.1	187.0	25.7	37.7	19.7	17.4	2.5	3.5	25.6	39.3	1.3	7.7
			*p* = 0.3339		*p* < 0.0001		*p* < 0.0001		*p* = 0.0886		*p* < 0.0001		*p* = 0.0096		*p* < 0.0001		*p* = 0.1940	
ICU EUR/m/CU	<900	358	82.1	204.4	170.2	204.0	113.5	176.7	28.1	41.7	13.2	16.5	2.4	3.2	12.5	32.3	0.6	5.9
	[900–1340]	421	89.3	187.8	178.1	186.0	125.6	159.9	25.4	36.2	12.8	15.7	2.1	2.8	11.3	25.1	0.7	5.6
	[1340–1850]	471	123.2	243.1	243.0	230.1	171.1	204.5	33.4	44.7	16.7	17.3	3.1	3.6	17.5	31.4	1.3	7.8
	>=1850	708	122.3	230.5	240.6	216.8	169.6	190.2	30.2	48.8	16.7	16.6	2.7	3.8	19.9	33.3	1.4	8.0
	Unknown	163	132.1	243.3	218.4	219.3	159.8	188.9	27.6	32.9	15.4	18.1	2.2	2.6	12.2	25.1	1.1	6.8
			*p* = 0.1287		*p* < 0.0001		*p* = 0.0010		*p* = 0.2076		*p* = 0.0152		*p* = 0.0059		*p* = 0.0029		*p* = 0.2525	
SPC	Low	499	75.4	171.9	160.0	179.7	108.3	156.0	30.6	47.8	10.4	14.2	2.7	3.5	7.3	18.1	0.8	7.0
	Medium	435	142.1	262.1	226.4	238.7	168.7	209.2	21.9	35.0	16.9	18.3	2.0	2.9	15.4	33.8	1.4	8.3
	High	349	149.3	251.5	265.8	222.5	189.1	200.0	34.5	41.1	17.4	16.1	2.9	3.1	21.0	30.9	0.9	5.6
	Not Working	837	91.7	206.7	215.9	209.0	146.1	180.0	30.6	43.6	16.3	17.2	2.7	3.6	19.2	35.0	1.0	6.6
			*p* = 0.0009		*p* < 0.0001		*p* < 0.0001		*p* = 0.0026		*p* < 0.0001		*p* = 0.0078		*p* < 0.0001		*p* = 0.5746	
Education BMI (kg/m^2^)	<12 y	803	69.7	175.4	174.1	183.3	116.0	152.8	25.9	39.4	13.0	15.4	2.3	3.2	16.2	35.3	0.6	5.1
	12 y	441	113.4	225.0	196.0	222.1	140.3	197.7	28.2	41.2	14.1	17.0	2.5	3.5	9.6	21.8	1.2	9.4
	12–16 y	448	144.2	240.6	265.4	227.7	189.5	202.7	37.3	56.6	17.7	18.5	3.1	3.9	15.8	26.1	2.0	9.2
	≥16 y	428	169.2	283.1	273.6	241.5	200.6	219.0	31.8	36.8	18.8	17.2	2.7	3.0	18.6	28.5	1.0	5.6
			*p* < 0.0001		*p* < 0.0001		*p* < 0.0001		*p* = 0.0571		*p* = 0.0001		*p* = 0.1230		*p* < 0.0001		*p* = 0.0708	
	Thin	53	192.4	295.2	243.8	234.5	182.7	223.5	32.3	35.4	17.4	16.6	3.7	4.2	6.2	12.4	1.4	7.0
	Normal	993	127.1	232.8	223.5	221.0	161.5	197.3	30.5	44.5	15.1	16.4	2.7	3.6	12.5	23.3	1.2	6.8
	Overweight	718	94.7	217.9	211.7	216.5	144.4	181.8	27.4	41.7	16.1	18.1	2.4	3.2	20.4	38.9	0.9	7.9
	Obesity	250	73.5	177.7	180.4	191.9	122.4	160.6	26.4	38.1	12.1	12.6	2.2	2.9	16.7	32.2	0.6	5.1
	Morbid obesity	102	41.7	116.2	131.2	113.6	73.7	100.0	35.6	52.6	11.8	18.1	2.4	3.1	6.9	21.2	0.6	5.1
			*p* < 0.0001		*p* < 0.0001		*p* < 0.0001		0.6749		0.0215		0.1368		0.0005		0.4028	
Children	All	1775	5.8	37.2	97.4	112.3	45.7	83.2	38.0	49.2	5.1	8.2	3.2	3.8	5.0	17.1	0.3	2.8

**Table 2 nutrients-16-01118-t002:** Distribution of total flavonoid intakes for children and adults—mean, standard deviation (SD), and percentiles.

		N	Total Flavonoids	SD	P5	P15	P25	P50	P75	P85	P95
Adults	All	2121	210.0	213.4	4.9	29.7	58.1	138.9	298.9	426.8	634.8
Gender	Men	887	205.5	195.1	4.9	35.1	62.2	142.8	308.3	395.1	566.1
	Women	1234	214.3	229.2	4.0	27.1	53.8	131.0	292.7	440.6	691.5
Age *	18–44 y.o.	783	176.9	190.5	3.0	23.0	48.0	111.8	231.3	355.5	577.6
	45–64 y.o.	827	232.0	230.1	3.9	31.0	62.0	156.2	350.1	440.6	640.5
	65–79 y.o.	511	250.8	221.0	17.8	48.6	86.3	177.8	364.3	471.4	710.1
ICU *	<900 EUR/m/CU	358	170.2	204.0	1.2	13.4	35.7	101.6	206.2	366.5	555.0
	[900–1340] EUR/m/CU	421	178.1	186.0	2.7	17.7	44.5	109.6	266.9	373.6	551.7
	[1340–1850] EUR/m/CU	471	243.0	230.1	8.7	45.0	72.1	187.2	362.2	439.5	711.4
	>=1850 EUR/m/CU	708	240.6	216.9	18.4	54.8	83.4	174.5	342.5	454.4	652.6
	Unknown	163	218.4	219.3	6.9	34.5	49.6	133.3	328.2	519.5	653.4
SPC *	Low	499	160.0	179.7	1.5	15.2	41.5	98.8	217.6	299.8	546.8
	Medium	435	226.4	238.7	3.7	17.6	54.8	138.9	357.5	440.6	757.1
	High	349	265.8	222.5	17.5	59.8	100.2	206.6	390.1	522.5	653.4
	Not Working	837	215.9	209.0	12.4	39.3	66.7	141.0	306.9	423.8	651.3
Education *	<12 y	803	174.1	183.3	3.6	20.6	48.6	107.0	241.2	364.3	543.0
	12 y	441	196.0	222.1	1.6	13.8	41.6	120.1	271.6	387.1	689.0
	12–16 y	448	265.4	227.7	16.5	56.4	88.9	207.6	381.4	466.1	699.8
	≥16 y	428	273.6	241.5	15.8	65.5	107.4	208.3	368.6	526.8	654.7
BMI *	Thin	53	243.8	234.5	11.6	39.0	70.6	160.4	496.0	566.1	626.2
	Normal	993	223.5	221.0	8.3	36.4	62.5	145.3	325.1	447.3	654.7
	Overweight	718	211.7	216.5	3.7	31.8	59.0	138.9	311.3	397.9	621.0
	Obesity	250	180.4	191.9	2.1	17.8	34.5	129.7	271.6	335.9	607.9
	Morbid obesity	102	131.2	113.6	1.1	15.8	51.5	107.0	182.5	254.9	324.8
Children	All	1775	97.4	112.3	1.6	18.1	33.7	70.0	127.0	169.5	269.3

* Mean total flavonoids intake was significantly different regarding age, ICU, SPC, education, and BMI.

**Table 3 nutrients-16-01118-t003:** Flavonoid intake of adults (ages >= 18 y) by food group (sorted by total flavonoids). Data for food groups containing flavonoids only.

	Quantity (g/d)	Total Flavonoids (mg/d)	Flavan-3-ols (mg/d)	Flavanones (mg/d)	Flavonols (mg/d)	Flavones (mg/d)	Anthocyanins (mg/d)	Isoflavones (mg/d)
	Mean (SD)	Mean (SD)	Mean (SD)	Mean (SD)	Mean (SD)	Mean (SD)	Mean (SD)	Mean (SD)
Brewed tea	106.8 (220.8)	82.8 (171.4)	77.2 (159.9)	0.0 (0.0)	5.7 (11.7)	0.0 (0.0)	0.0 (0.0)	0.0 (0.0)
Fruits, nuts, and seeds	147.1 (139.8)	46.4 (69.8)	29.7 (48.8)	7.8 (22.9)	2.0 (3.3)	0.3 (0.7)	6.6 (21.2)	0.0 (0.0)
Alcoholic beverages	128.4 (247.4)	25.9 (64.5)	15.3 (37.7)	0.2 (0.4)	1.8 (4.4)	0.0 (0.0)	8.6 (21.9)	0.0 (0.0)
Fruit and vegetable juices	66.0 (102.1)	23.2 (40.5)	0.0 (0.0)	21.0 (37.4)	0.3 (0.6)	1.9 (3.0)	0.0 (0.0)	0.0 (0.0)
Chocolate confectionery	6.7 (14.8)	14.6 (43.5)	14.3 (42.3)	0.0 (0.0)	0.4 (1.2)	0.0 (0.0)	0.0 (0.0)	0.0 (0.0)
Dairy products and substitutes	251.6 (217.6)	7.9 (32.7)	6.9 (32.0)	0.0 (0.0)	0.1 (0.1)	0.0 (0.0)	0.0 (0.0)	0.9 (6.6)
Vegetables	136.2 (113.8)	4.5 (9.6)	0.1 (0.4)	0.2 (0.5)	4.1 (9.4)	0.1 (0.3)	0.0 (0.1)	0.0 (0.0)
Coffee and herbal teas	340.0 (316.5)	3.1 (23.2)	3.1 (23.2)	0.0 (0.0)	0.0 (0.0)	0.0 (0.0)	0.0 (0.0)	0.0 (0.0)
Legumes	7.9 (25.3)	0.5 (1.9)	0.1 (0.4)	0.0 (0.0)	0.4 (1.7)	0.0 (0.1)	0.0 (0.0)	0.0 (0.0)
Potatoes and other tubers	55.8 (80.6)	0.4 (0.8)	0.0 (0.0)	0.0 (0.0)	0.4 (0.8)	0.0 (0.0)	0.0 (0.0)	0.0 (0.0)
Condiments, spices, and sauces	24.3 (29.1)	0.3 (1.4)	0.0 (0.0)	0.0 (0.0)	0.0 (0.1)	0.2 (1.4)	0.0 (0.0)	0.0 (0.0)
Sugar, honey, jam, syrup	19.4 (22.3)	0.2 (0.4)	0.1 (0.2)	0.0 (0.0)	0.0 (0.0)	0.0 (0.0)	0.1 (0.2)	0.0 (0.0)
Meat, meat products, substitutes	106.4 (75.0)	0.1 (1.2)	0.0 (0.0)	0.0 (0.0)	0.0 (0.0)	0.0 (0.0)	0.0 (0.0)	0.1 (1.2)
Carbonated soft isotonic drinks	99.8 (216.9)	0.0 (0.1)	0.0 (0.1)	0.0 (0.0)	0.0 (0.0)	0.0 (0.0)	0.0 (0.0)	0.0 (0.0)
Cereals and cereal products	190.0 (118.8)	0.0 (0.1)	0.0 (0.0)	0.0 (0.0)	0.0 (0.1)	0.0 (0.1)	0.0 (0.0)	0.0 (0.0)
Fats and oils	14.50 (13.3)	0.0 (0.0)	0.0 (0.0)	0.0 (0.0)	0.0 (0.0)	0.0 (0.0)	0.0 (0.0)	0.0 (0.0)
Miscellaneous	1.6 (13.8)	0.0 (0.1)	0.0 (0.0)	0.0 (0.0)	0.0 (0.0)	0.0 (0.0)	0.0 (0.0)	0.0 (0.1)
**TOTAL**	**1703**	**210.0**	**146.9**	**29.2**	**15.0**	**2.6**	**15.3**	**1.0**

**Table 4 nutrients-16-01118-t004:** Flavonoid intakes of children (ages 4–17 y) by food group (sorted by total flavonoids). Data for food groups containing flavonoids only.

	Quantity (g/d)	Total Flavonoids (mg/d)	Flavan-3-ols (mg/d)	Flavanones (mg/d)	Flavonols (mg/d)	Flavones (mg/d)	Anthocyanins (mg/d)	Isoflavonoids (mg/d)
	Mean (SD)	Mean (SD)	Mean (SD)	Mean (SD)	Mean (SD)	Mean (SD)	Mean (SD)	Mean (SD)
Fruit and vegetable juices	103.5 (123.1)	38.2 (50.9)	0.0 (0.0)	34.9 (47.2)	0.4 (0.7)	2.9 (3.7)	0.0 (0.0)	0.0 (0.0)
Fruits, nuts, and seeds	110.6 (99.4)	26.3 (44.4)	17.0 (29.0)	3.0 (11.1)	1.1 (1.9)	0.2 (0.5)	5.0 (17.0)	0.0 (0.0)
Dairy products and substitutes	364.8 (228.1)	12.6 (22.5)	12.2 (22.2)	0.0 (0.0)	0.1 (0.2)	0.0 (0.0)	0.0 (0.0)	0.3 (2.5)
Chocolate confectionery	13.0 (21.5)	12.3 (72.3)	12.0 (70.6)	0.0 (0.0)	0.3 (1.9)	0.0 (0.0)	0.0 (0.0)	0.0 (0.0)
Brewed tea	5.8 (37.2)	4.4 (28.6)	4.1 (26.7)	0.0 (0.0)	0.3 (1.9)	0.0 (0.0)	0.0 (0.0)	0.0 (0.0)
Vegetables	79.7 (73.0)	2.6 (6.7)	0.0 (0.2)	0.2 (0.4)	2.3 (6.5)	0.1 (0.1)	0.0 (0.2)	0.0 (0.0)
Legumes	4.7 (15.7)	0.3 (1.2)	0.1 (0.2)	0.0 (0.0)	0.2 (1.1)	0.0 (0.0)	0.0 (0.0)	0.0 (0.0)
Potatoes and other tubers	52.7 (55.3)	0.3 (0.4)	0.0 (0.0)	0.0 (0.0)	0.3 (0.4)	0.0 (0.0)	0.0 (0.0)	0.0 (0.0)
Coffee and herbal teas	11.9 (57.2)	0.2 (2.7)	0.2 (2.7)	0.0 (0.0)	0.0 (0.0)	0.0 (0.0)	0.0 (0.0)	0.0 (0.0)
Condiments, spices, and sauces	20.3 (28.0)	0.1 (0.3)	0.0 (0.0)	0.0 (0.0)	0.0 (0.0)	0.1 (0.3)	0.0 (0.0)	0.0 (0.0)
Sugar, honey, jam, syrup	7.1 (12.9)	0.1 (0.2)	0.0 (0.1)	0.0 (0.0)	0.0 (0.0)	0.0 (0.0)	0.0 (0.1)	0.0 (0.0)
Meat, meat products, and substitutes	85.5 (54.8)	0.1 (1.0)	0.0 (0.0)	0.0 (0.0)	0.0 (0.0)	0.0 (0.0)	0.0 (0.0)	0.1 (1.0)
Carbonated soft isotonic drinks	121.0 (176.7)	0.0 (0.0)	0.0 (0.0)	0.0 (0.0)	0.0 (0.0)	0.0 (0.0)	0.0 (0.0)	0.0 (0.0)
Miscellaneous	1.2 (10.7)	0.0 (0.4)	0.0 (0.0)	0.0 (0.0)	0.0 (0.0)	0.0 (0.0)	0.0 (0.0)	0.0 (0.4)
Alcoholic beverages	2.0 (20.0)	0.0 (0.1)	0.0 (0.1)	0.0 (0.0)	0.0 (0.0)	0.0 (0.0)	0.0 (0.0)	0.0 (0.0)
Cereals and cereal products	158.1 (103.8)	0.0 (0.0)	0.0 (0.0)	0.0 (0.0)	0.0 (0.0)	0.0 (0.0)	0.0 (0.0)	0.0 (0.0)
Fats and oils	8.4 (8.2)	0.0 (0.0)	0.0 (0.0)	0.0 (0.0)	0.0 (0.0)	0.0 (0.0)	0.0 (0.0)	0.0 (0.0)
**TOTAL**	**1150**	**97.4**	**45.7**	**38.0**	**5.1**	**3.2**	**5.0**	**0.3**

**Table 5 nutrients-16-01118-t005:** Surveyed mean flavonoids intake (mg/d) according to the tea consumption status. Data for adults only.

ADULTS	Tea Non-Consumers (N = 1363)	Tea Consumers (N = 758)	*p*-Value (Unadjusted)	*p*-Value Adjusted **
Total flavonoids (mg)	128.0	385.4	<0.0001	<0.0001
Flavan-3-ols (mg)	69.4	312.6	<0.0001	<0.0001
Flavanones (mg)	29.5	28.7	0.7349	0.7696
Flavonols (mg)	9.5	26.7	<0.0001	<0.0001
Flavones (mg)	2.6	2.5	0.6157	0.8952
Anthocyanins (mg)	16.3	13.4	0.1535	0.1095
Isoflavones (mg)	0.7	1.6	0.0123	0.2172

All measures are the average of 2 or 3 day recalls. ** Adjusted for sex, age, ICU, SPC, education, and energy intake; least squared means not reported.

## Data Availability

All databases analyzed are publicly available and can be downloaded from agency websites in France and in the US. The INCA 3 dataset is available here: https://www.data.gouv.fr/en/datasets/donnees-de-consommations-et-habitudes-alimentaires-de-letude-inca-3/ accessed on 10 June 2020. Flavonoids data are available from http://phenol-explorer.eu/ accessed on 6 June 2023 and the USDA website: https://www.ars.usda.gov/northeast-area/beltsville-md-bhnrc/beltsville-human-nutrition-research-center/methods-and-application-of-food-composition-laboratory/mafcl-site-pages/flavonoids/ accessed on 6 June 2023. Codes for matching of datasets will be made available upon reasonable request.

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
