# Peer review of "Dietary Flavonoid Intakes in France Are Linked to Brewed Tea Consumption and to Socioeconomic Status: Analyses of the Third French Individual and National Food Consumption (INCA3) Survey for Children and Adults"

_nutrients, 2024, doi:10.3390/nu16081118_

Round 1
Reviewer 1 Report
Comments and Suggestions for Authors
The present paper is an original research regarding dietary flavonoid intakes in France performed for the first time by specialists from MS Nutrition and Center for Public Health Nutrition, University of Washington. By evaluating the flavonoid intakes of the adult and infant population in France by a statistical significant sample of over 2100 adults and over 1700 children using a self reported 24h dietary recall and a 3 days food journal with a consistent research methodology, mostly in comparison with the US population. The purpose of the study is matching a knowledge gap on the topic, that could (1) provide data regarding the flavonoid intake in the general population. in order fuel future research on topic.
The authors have done an excellent job, however, some issues can be improved.
1. Title
" Dietary flavonoid intakes in France are linked to brewed tea 2 consumption and to socioeconomic status: Analyses of INCA 3 3 data for children and adults.”
· “The expression of the title mightgive a indicative of the methodology of the study– Dietary flavonoid intakes in France are linked to brewed tea consumption and to socioeconomic status: Analyses of INCA 3 3 data for children and adults”
· …..based on food questionares …… at author's latitude.
2. Keywords: “Flavonoids; flavan-3-ols; brewed tea; demographics; socioeconomic status; France; 27 INCA 3” might include, “intake evaluation” or similar to point the research tool
3. Abstract
- Not very clear what INCA 3 means at this poin
- Please review the English language correctness and character number
4. References
- Most of the references are actual.
- Please check the correctness of the format
1. Introduction
· Line 61, please add the correct reference (description, city, country) for the (NHANES 2007-10), French PhenolExplorer, USDA’s Expanded Flavonoid Database as much as possibile in text.
· More introductory data on available research on specific health benefits for certain flavonoids briefly presented would be welcomed (line 35 or Line 49)
5. Materials and methods
· A study graphical diagram is recommended for a better understanding of the research.
· Line 85 - Please mention the city and country of the database resource used.
· Please explain what SURVEYREG, SURVEY-147 MEANS and SURVEYFREQ procedures mean.
· Line 1 page 16 – explanation of Tble 1 might be set before the table for clarity
6. Results
· Table 1 – hard to follow with the text might be fitted as supplementary or splitted
· Where possible, a graphical representation of results might be clearer
· Tables –maintaining the same number of decimals is better viewed
Discussions
The main comparison is with USA flavonoid consumption. Are there any comparisons with other countries/regions/EU regions?
7. Conclusions
A study perspective might be added.
The abbreviation section, even if optional, would improve the clarity of the reading.

Author Response
Comments and Suggestions for Authors
The present paper is an original research regarding dietary flavonoid intakes in France performed for the first time by specialists from MS Nutrition and Center for Public Health Nutrition, University of Washington. By evaluating the flavonoid intakes of the adult and infant population in France by a statistical significant sample of over 2100 adults and over 1700 children using a self reported 24h dietary recall and a 3 days food journal with a consistent research methodology, mostly in comparison with the US population. The purpose of the study is matching a knowledge gap on the topic, that could (1) provide data regarding the flavonoid intake in the general population. in order fuel future research on topic.
The authors have done an excellent job, however, some issues can be improved.
Thank you for the time spent in reviewing this article. We appreciate the constructive comments.
- Title
" Dietary flavonoid intakes in France are linked to brewed tea 2 consumption and to socioeconomic status: Analyses of INCA 3 3 data for children and adults.”
- “The expression of the title mightgive a indicative of the methodology of the study– Dietary flavonoid intakes in France are linked to brewed tea consumption and to socioeconomic status: Analyses of INCA 3 3 data for children and adults”
- …..based on food questionares …… at author's latitude.
Response: We have spelled out the INCA 3 survey for greater clarity:
The Third French Individual and National Food Consumption (INCA3) Survey
The new title is: Dietary flavonoid intakes in France are linked to brewed tea consumption and to socioeconomic status: Analyses of The Third French Individual and National Food Consumption (INCA3) survey for children and adults
- Keywords: “Flavonoids; flavan-3-ols; brewed tea; demographics; socioeconomic status; France; 27 INCA 3” might include, “intake evaluation” or similar to point the research tool
Response: We now use the formal title ”The Third French Individual and National Food Consumption (INCA3) Survey” that points directly to the research tool.
- Abstract
- Not very clear what INCA 3 means at this point
Response: We now have the full name and the abbreviation in the Abstract. The Third French Individual and National Food Consumption (INCA3) Survey .
- Please review the English language correctness and character number
Response The abstract is 217 words
- References
- Most of the references are actual.
- Please check the correctness of the format
- Introduction
- Line 61, please add the correct reference (description, city, country) for the (NHANES 2007-10), French PhenolExplorer, USDA’s Expanded Flavonoid Database as much as possible in text.
Response Links to databases used on the study (INCA 3, Phenol Explorer , USDA flavonoid database) are provided in the Data availability section. This is made clear in test on line 65. The references are to past literature that has made use of these data. For example, Sebastian et al used early flavonoid data linked to the NHANES 2007-10 database. The 3 references to PhenolExplorer (references 11-13) are to articles and not the database.
The Phenol-Explorer website is indicated in the “Data Availability Statement” of our manuscript. cited. The USDA flavonoids database is also indicated in the “Data Availability Statement”. The references to the USDA’s expanded flavonoids database (ref 14-16) are reports related to the database available in USDA website.
More introductory data on available research on specific health benefits for certain flavonoids briefly presented would be welcomed (line 35 or Line 49)
Response A sentence has been added to the introduction: “Flavonoids, and particularly flavan-3-ols and anthocyanidins subclasses, have been associated with a reduction in the risk of cardiovascular diseases [4].”
- Materials and methods
- A study graphical diagram is recommended for a better understanding of the research.
Response: A diagram has been integrated in the beginning of the material and methods section
- Line 85 - Please mention the city and country of the database resource used.
Response: Line 85 refers to a software. A reference has been added: Slimani, N.; Casagrande, C.; Nicolas, G.; Freisling, H.; Huybrechts, I.; Ocké, M.C.; Niekerk, E.M.; Van Rossum, C.; Bellemans, M.; De Maeyer, M.; et al. The standardized computerized 24-h dietary recall method EPIC-Soft adapted for pan-European dietary monitoring. Eur. J. Clin. Nutr. 2011, 65, S5–S15.
Please explain what SURVEYREG, SURVEY-147 MEANS and SURVEYFREQ procedures mean.
Response: We have rephrased the paragraph.
- Line 1 page 16 – explanation of Tble 1 might be set before the table for clarity
Response: All the text which describes Table 1 has been placed before the table.
- Results
- Table 1 – hard to follow with the text might be fitted as supplementary or splitted
Response: Table 1 was reformatted and is much easier to follow. The table provides a direct comparison to the US data which were presented in a similar format.
Where possible, a graphical representation of results might be clearer
Response: three graphical (2 in results section and 1 in material and methods) representations were added.
- Tables –maintaining the same number of decimals is better viewed
Response Following reviewer comment, we kept 1 decimal for all values in all tables. Including supplemental tables
Discussions
The main comparison is with USA flavonoid consumption. Are there any comparisons with other countries/regions/EU regions?
Response: There are no data on flavonoids from any other country that we are aware of. To do this, nutrient composition databases must be merged with flavonoid databases and only two of those are in existence – USDA and PhenolExplorer. This is the first analysis of flavonoid content in the Third French Individual and National Food Consumption Survey
- Conclusions
Response: The conclusion section was rewritten to provide additional perspective
The abbreviation section, even if optional, would improve the clarity of the reading.
Response: An abbreviation section was added at the end. .
Reviewer 2 Report
Comments and Suggestions for Authors
Review of manuscript ref. nutrients-2889480
Title: Dietary flavonoid intakes in France are linked to brewed tea consumption and to socioeconomic status: Analyses of INCA 3 data for children and adults
Authors: Florent Vieux , Matthieu Maillot , Adam Drewnowski
General comment: the study reports the results of a theoretical calculation of flavonoid consumption after a three-day survey of food intake in French children and adults. Based on a nationally representative INCA 3 dietary survey of the French population conducted by the French Agency for Food, Environmental and Occupational Health & Safety (ANSES) in France between 2014 and 2015, the dietary intake assessment reveals that in adults, most flavonoids were provided by flavan-3-ols of which tea is the main source. Besides, both tea consumption and flavonoid intakes were associated with higher socio-economic status. The objective and design of the observational study are sound and simple, evaluation based on two main data bases is adequate, and results are very interesting for dietary and nutritional purposes. The major concern is the complete overlook of cocoa and derivatives when cocoa is the product with the highest amount of flavanols and chocolates containing over 70 % of cocoa are a significant source of flavanols. Although it is true that cocoa soluble products are more consumed in other European countries such as Great Britain or Spain, it seems strange that these foodstuffs are not even mentioned in the whole text. Some specific comments are detailed below:
Specific comments:
1) Figure 1, panel b; it should say flavanols or flavan-3-ols in the legend, instead of flavonols.
2) Tables 3 and 4; it seems that some names of food groups are incomplete and should be double checked: Condiments, spices, sauces and y… and Meat, meat products and substitu…
3) Since flavan-3-ols accounted for 39% of the flavonoid intake, cocoa and derivatives (mostly chocolate) are a main source of flavan-3-ols, and French people are reasonably devoted of such products, in my opinion cocoa and chocolate should deserve a category of food group or, at least, be mentioned in any of the food groups stated in the tables. Besides, they should also be mentioned in the manuscript.
4) Although the study is focused on flavonoids, since according to table 4 consumption of coffee in France doubles that of tea, it would be very interesting (in further studies) to address intake of other common polyphenols such as hydroxycinnamic acids, particularly chlorogenic acid, a highly bioactive compound from coffee and berries.
5) Lines 168-169; it should say flavan-3-ols or flavanols.
Comments on the Quality of English LanguageNo further comments
Author Response
Comments and Suggestions for Authors
General comment: the study reports the results of a theoretical calculation of flavonoid consumption after a three-day survey of food intake in French children and adults. Based on a nationally representative INCA 3 dietary survey of the French population conducted by the French Agency for Food, Environmental and Occupational Health & Safety (ANSES) in France between 2014 and 2015, the dietary intake assessment reveals that in adults, most flavonoids were provided by flavan-3-ols of which tea is the main source. Besides, both tea consumption and flavonoid intakes were associated with higher socio-economic status. The objective and design of the observational study are sound and simple, evaluation based on two main data bases is adequate, and results are very interesting for dietary and nutritional purposes. The major concern is the complete overlook of cocoa and derivatives when cocoa is the product with the highest amount of flavanols and chocolates containing over 70 % of cocoa are a significant source of flavanols. Although it is true that cocoa soluble products are more consumed in other European countries such as Great Britain or Spain, it seems strange that these foodstuffs are not even mentioned in the whole text. Some specific comments are detailed below:
Thank you for the time spent in this review. We better detailed the “Sugar and confectionery” food group in order to highlight the contribution of chocolate.
Specific comments:
- Figure 1, panel b; it should say flavanols or flavan-3-ols in the legend, instead of flavonols.
Thank you, we modified.
- Tables 3 and 4; it seems that some names of food groups are incomplete and should be double checked: Condiments, spices, sauces and y… and Meat, meat products and substitu…
Response :Thank you, we modified the name of those food categories
- Since flavan-3-ols accounted for 39% of the flavonoid intake, cocoa and derivatives (mostly chocolate) are a main source of flavan-3-ols, and French people are reasonably devoted of such products, in my opinion cocoa and chocolate should deserve a category of food group or, at least, be mentioned in any of the food groups stated in the tables. Besides, they should also be mentioned in the manuscript.
Response: Thank you for this comment. That is absolutely correct. We separated out chocolate confectionery from the broader “Sugar and confectionery” food group. As expected, “Sugar, honey, jam, syrup” and “Confectionery non chocolate” contained very little flavonoids and were dropped (data provided in supplemental Table 1). Chocolate confectionery was indeed the main contributor to flavonoids and is now features as a separate category.
- Although the study is focused on flavonoids, since according to table 4 consumption of coffee in France doubles that of tea, it would be very interesting (in further studies) to address intake of other common polyphenols such as hydroxycinnamic acids, particularly chlorogenic acid, a highly bioactive compound from coffee and berries.
Response: Other polyphenols are now mentioned in the Discussion section
5) Lines 168-169; it should say flavan-3-ols or flavanols.
Response: Thank you, we modified accordingly.